# The Safety and Toxicity of Phage Therapy: A Review of Animal and Clinical Studies

**DOI:** 10.3390/v13071268

**Published:** 2021-06-29

**Authors:** Dan Liu, Jonas D. Van Belleghem, Christiaan R. de Vries, Elizabeth Burgener, Qingquan Chen, Robert Manasherob, Jenny R. Aronson, Derek F. Amanatullah, Pranita D. Tamma, Gina A. Suh

**Affiliations:** 1Department of Burn, School of Medicine, Shanghai Jiao Tong University, Shanghai 200025, China; k_liudan@sjtu.edu.cn; 2Division of Infectious Diseases and Geographic Medicine, Department of Medicine, School of Medicine, Stanford University, Stanford, CA 94305, USA; van.belleghem.jonas@gmail.com (J.D.V.B.); devries2@stanford.edu (C.R.d.V.); kevinqc@stanford.edu (Q.C.); aronsonj@stanford.edu (J.R.A.); 3Center for Excellence in Pulmonary Biology, Department of Pediatrics, Stanford University, Stanford, CA 94305, USA; eburgener@stanford.edu; 4Department of Orthopaedic Surgery, School of Medicine, Stanford University, Stanford, CA 94305, USA; robertm1@stanford.edu (R.M.); dfa@stanford.edu (D.F.A.); 5Division of Pediatric Infectious Diseases, Department of Pediatrics, School of Medicine, Johns Hopkins University, Baltimore, MD 21287, USA; ptamma1@jhmi.edu; 6Division of Infectious Diseases, Department of Medicine, Mayo Clinic, Rochester, MN 55905, USA

**Keywords:** phage therapy, clinical trials, animal models, safety and toxicity, immune activation

## Abstract

Increasing rates of infection by antibiotic resistant bacteria have led to a resurgence of interest in bacteriophage (phage) therapy. Several phage therapy studies in animals and humans have been completed over the last two decades. We conducted a systematic review of safety and toxicity data associated with phage therapy in both animals and humans reported in English language publications from 2008–2021. Overall, 69 publications met our eligibility criteria including 20 animal studies, 35 clinical case reports or case series, and 14 clinical trials. After summarizing safety and toxicity data from these publications, we discuss potential approaches to optimize safety and toxicity monitoring with the therapeutic use of phage moving forward. In our systematic review of the literature, we found some adverse events associated with phage therapy, but serious events were extremely rare. Comprehensive and standardized reporting of potential toxicities associated with phage therapy has generally been lacking in the published literature. Structured safety and tolerability endpoints are necessary when phages are administered as anti-infective therapeutics.

## 1. Introduction

Antibiotic resistance remains an ongoing global threat. The failure to implement widespread stewardship over these precious resources, the resulting spread of antibiotic resistance, and an under-resourced antibiotic pipeline portend the coming of a “post-antibiotic era” [1]. Lytic bacteriophages (phages) have been known to be a potential antibacterial agent for over a century since their first formal discovery and application as a treatment against human bacterial infections in the 1920s. Though subsequent success of antibiotics had quelled investigations into phage as potential anti-infectives, increasing antibiotic resistance has hastened the reemergence of interest in phage therapy [2,3,4].

Despite attractive advantages, including widespread prevalence, activity against multidrug-resistant (MDR) bacteria, high specificity, and activity against biofilms [5,6], phage therapy is still not considered a mainstream treatment due to several obstacles. First, the lack of reliable data regarding its safety and efficacy in clinical settings [7,8,9]. Second, appropriate regulatory guidelines specific to phage therapeutics have not been developed [10,11]. Finally, the pharmaceutical and biotech industries have not yet developed economical and scalable production processes for widespread adoption of phage therapy. While much progress has been made, many questions remain [2,12,13,14], such as the safety and toxicity of phage preparations.

To ensure confidence in the use of phages as anti-infectives, it is critical that basic safety issues of phage therapy are established [8,15,16]. Here, we focus on the safety of phage therapy in clinical trials, case reports, and in vivo animal model studiessince 2008. We review the current safety assessments in phage therapy research and reported safety issues. We found that there were gaps and a lack of systematic and detailed descriptions regarding safety issues. However, there were no major adverse events reported. We then discuss the potential safety issues of phage therapy, including the effects of phage on the human body, human flora, immune system, and the release of endotoxin caused by lysis of the bacteria. Furthermore, bacterial residues in phage preparations and possible chemical components in the purification process can influence the safety of phage therapy. Finally, we discuss the challenges and opportunities to optimize methods of phage therapy safety monitoring and quality control to enhance implementationstrategies.

There is significant contribution to this literature from Europe with 30% (6/20) of animal studies, 43% (15/35) of case reports, and 50% (7/14) of clinical trials published out of Europe.

## 2. Materials and Methods

In this section, we focus on the safety of phage therapy through a review of both clinical trials and in vivo animal studies. We have limited these efforts to reports published in English language journals between 1 January 2008 and 28 March 2021 available on PubMed. We excluded the following: (1) non-vertebrate animal studies, (2) reports of phage use other than for anti-infective purposes, (3) phage lysin investigations, and (4) investigations into non-lytic phage. We identified 20 animal studies, 35 case reports/series, and 14 clinical trials that met our eligibility criteria (Figure 1).

The search terms “phage therapy” or “bacteriophage therapy” were used in PubMed. The study types were defined as “Clinical Trials”, “Case Reports” and “Animal Studies”, respectively. The total of 14 clinical trials, 35 case reports, and 20 animal studies were included and discussed in this review.

## 3. Results

### 3.1. Animal Studies

Of the 20 animal studies, four studies focused on safety (Table 1). Dufour et al. showed that phages were capable of modulating the immune response in a phage-specific manner, unknown biological significances but without any adverse observable clinical ones. This murine acute pneumonia model was initiated by intranasal instillation of two *Escherichia coli* strains (536 and LM33) and treated by two phages (536_P1 and LM33_P1; intranasal) with antibiotics (ceftriaxone, cefoxitin, or imipenem-cilastatin) as a comparator. The complete blood counts (CBC), lung edema, cytokine level, bacterial, and phage counts were determined. Phage and antibiotics displayed similar endpoints, but phage decreased the bacterial load and corrected blood cell count abnormalities at a more rapid rate. The rapid lysis of bacteria by phages did not increase the innate inflammatory response compared to antibiotics. Meanwhile, phage 536_P1 promoted a weak increase in antiviral cytokines IFN-γ and IL-12 in the lungs, which was not observed in infected animals [15]. A long-term study (20 days) by Drilling and colleagues investigated the safety of topical sinonasal flushes with phage cocktail NOV012 (P68 and K710) against *Staphylococcus aureus* in the same model. General wellbeing, mucosal structural changes, and inflammatory load were assessed. With no inflammatory infiltration or tissue damage within the sinus mucosa observed, the application of NOV012 was found to be safe [16]. Fong and colleagues assessed the safety of a *Pseudomonas aeruginosa* phage cocktail (CT-PA) in a sinusitis sheep model. After a 7-day biofilm formation period, sheep received frontal trephine flushes of CT-PA twice-daily for 1 week. Blood and fecal samples were collected. Histopathology of frontal sinus, lung, heart, liver, spleen, and kidney tissue was performed. Phages were detected consistently in feces and sporadically in blood and organs. Sinus cilia were visualized using SEM. The authors showed that CT-PA reduced the biofilm biomass significantly. No safety concerns of tissues were noted [17].

Sixteen efficacy studies evaluated safety measures in some capacity, including general health, physical examination, hematology, organ function, or immune response (Appendix A). In contrast to Dufour’s conclusions that phage therapy did not induce an inflammatory response, phage therapy for endocarditis in rats induced by *P. aeruginosa* showed that phage therapy, but not ciprofloxacin, correlated with significantly increased plasma levels of IL-1β and IL-6. Because ciprofloxacin is not bacteriolytic, the increase in IL-1β and IL-6 levels was considered to be related to phage-induced bacterial lysis [18]. Two animal studies showed that treatment with phage leads to increased anti-phage antibody titers. For example, a 170-fold and 50-fold increase in IgG and IgM titers against phage in mice *Vibrio parahaemolyticus* were observed [19,20]. Another study looking at *S. aureus* bacteremia in mice showed a 2500-fold and 100-fold increase of IgG and IgM, respectively, after intra-peritoneal (IP) phage administration [19,20]. No other adverse effects were reported in these two studies.
viruses-13-01268-t001_Table 1Table 1Safety monitoring in phage therapy studies.**Reference****Physical Exam****Adverse Events****Distribution****Laboratory Assessment****Immune Response****Inflammation****Ig****Production****Animal Studies****Hematology****Liver Function****Kidney Function****Electrolytes****Cell****Infiltration****Cytokine**Dufour et al., 2019 [15]









Fong et al., 2019 [17]









Drilling et al., 2017 [16]









Drilling et al., 2014 [21]









ReferenceSubjective DataPhysical ExamAdverse EventImagingDistributionLab Exam.Immune ResponseSystem InflammationIg ProductionCase ReportsHematologyLiver FunctionKidney FunctionElectrolyteCRP or ESRCytokineLebeaux et al., 2021 [22]











Ferry et al., 2020 [23]











Bao et al., 2020 [24]











Cano et al., 2020 [25]











Rostkowska et al., 2020 [26]











Doub et al., 2020 [27]











Rubalskii et al., 2020 [28]











Gainey et al., 2020 [29]











Aslam et al., 2019 [30]











Nir-Paz et al., 2019 [31]











Tkhilaishvili et al., 2019 [32]











Onsea et al., 2019 [33]











Corbellino et al., 2019 [34]











Susan et al., 2019 [35]











Gilbey et al., 2019 [36]











Law et al., 2019 [37]











RM et al., 2019 [38]











Kuipers et al., 2019 [39]











Fish et al., 2018 [40]











Ferry et al., 2018 [41]











 Hoyle et al., 2018 [42]











Chan et al., 2018 [43]











Duplessis et al., 2019 [10] 











LaVergne et al., 2018 [44]











Ferry et al., 2018 [45]











Ujmajuridze et al., 2018 [46]











Schooley et al., 2017 [47]











Zhvania et al., 2017 [48]











Jennes et al., 2017 [49]











Fish et al., 2016 [50]











Fadlallah et al., 2015 [51]











Rose et al., 2014 [52]











Khawaldeh et al., 2011 [53]











Kvachadze et al., 2011 [54]











Letkiewicz et al., 2009 [55]











Clinical Trials











Leitner et al., 2020 [56]











Grubb et al., 2020 [57]











Fabijan et al., 2020 [58]











Ooi et al., 2019 [59]











Febvre et al., 2019 [60]











Gindin et al., 2018 [61]











McCallin et al., 2018 [62]











Sarker et al., 2017 [63]











McCallin et al., 2013 [64]











Sarker et al., 2012 [65]











Rhoads et al., 2009 [66]











Patrick et al., 2018 [67]











Sarker et al., 2016 [68]











Wright et al., 2009 [69]











Dark Blue = have values or result within article; Grey = not mentioned within article. “Subjective Data” includes feedback from healthy volunteers or patients taken during or after phage administration. “Physical Exam” data include vital signs and physical exam findings. “Phage Distribution” refers to presence of phage in blood or other organs besidessite of infection. “Adverse Event” refers to any reported adverse events, regardless of severity. “Imaging” refers to any imaging test, including ultrasound, x-ray, CT, MRI, etc. “Lab Exam” denotes clinical laboratory testing including hematology, liver and kidney function, and electrolytes. “Immune Response” refers to systemic inflammatory markers, such CRP and ESR, topical or circulating cytokine levels. “Ig Production” refers to testing for immunoglobulins in blood or feces.


### 3.2. Case Reports

Thirty-five case reports/series of phage therapy were published between 2008–2021 (Appendix A). Most involved the combined use of phages with antibiotics, targeting a variety of pathogens (Appendix A). The conditions treated included cystic fibrosis exacerbation, bone/joint infection, pneumonia, bacteremia, urinary tract infection (UTI), endocarditis, cardiothoracic surgery-related infections, aorto-cutaneous fistula, necrotizing pancreatitis, skin infection, brain infection, diabetic foot ulcers, corneal abscess, lung transplant-related infection, and intestinal infection. Twenty-seven cases included safety measures (Appendix A), including subjective symptom reporting, physical examination, hematologic measurements, liver function, kidney function, electrolytes, imaging, and adverse events. Some studies also included additional clinical markers such as erythrocyte sedimentation rate (ESR), C-reactive protein (CRP), cytokine levels, and anti-phage antibody production.

Among these 35 studies, a 72-year-old male with a chronic methicillin-resistant *S. aureus* prosthetic joint infection developed a reversible transaminitis after three intravenous (IV) doses of phage, prompting discontinuation of phage therapy. No other liver function derangement was seen, and the transaminitis was non-life threatening. The investigators hypothesized that underlying steatosis induced a dysregulated local cytokine response in the macrophages within the liver when challenged with large amounts of phages that needed to be cleared [27]. Another case report involved a 68-year-old diabetic patient with necrotizing pancreatitis with *Acinetobacter baumannii*. Two days following the IV phage administration, the patient’s vasopressor requirements abruptly increased, and phage therapy was temporarily withheld. It was subsequently demonstrated that the clinical deterioration was accompanied by a transient septic episode. Phage therapy was resumed about a week later and the patient’s condition improved [47]. For a two-year-old male with *P. aeruginosa* bacteremia, phage therapy was withheld due to anaphylaxis-related decompensation, which was attributed to progressive heart failure, although endotoxin release could not be excluded as a contributing factor. Shortly after phage therapy resumed, the patient had clinical improvement [10]. Another case series showed a patient with *P. aeruginosa* induced UTI experiencing sudden fever (38.5 °C) and chills on the third day of phage therapy, which were considered to be related to released endotoxins during *P. aeruginosa* lysis. The phage treatment was subsequently stopped. The body temperature normalized 48 h after changes were made to the antibiotic regimen [46]. Moreover, bacterial components and toxins such as endotoxin, could have the potential to induce these infusion-related reactions. For example, a 77-year-old male with a multidrug-resistant *A. baumannii* craniectomy site infection developed hypotension 115 min after the first dose of phage therapy. As this did not require vasopressors, phage treatment was continued [44]. Another case involving a 15-year-old patient with cystic fibrosis and a *Mycobacterium abscessus* infection was reported to feel sweaty and flushed but had no fever or changes on physical exam after IV phage administration. Otherwise phage therapy was well tolerated throughout without significant side effects [38].

### 3.3. Clinical Trials

The first investigation into the bioavailability of oral *E. coli* phage T4 in 2005 involving fifteen healthy humans did not identify any adverse events [70]. Since 2008 there have been 14 clinical trials of phage therapy (Appendix A) investigating a multitude of bacterial infections (Appendix A). Indications for phage therapy included endocarditis, sepsis, rhinosinusitis, UTI, venous leg ulcers, chronic otitis media, acute bacterial diarrhea, and burn wounds (Appendix A). All of these trials evaluated safety measures (Table 1). The safety endpoints reported by these trials included subjective data/symptom reporting, physical examination, hematologic measurements, liver function, kidney function, and electrolytes.

Among these trials, a double-blinded, placebo-controlled crossover trial was carried out, in which healthy adults consumed a commercial cocktail of *E. coli*-targeting bacteriophages for 28 days [60]. The gut microbiota and markers of intestinal and systemic inflammation were examined. There was only a small but significant decrease in circulating IL-4. Inflammatory markers (CRP, GM-CSF, IFNγ, IL-1α, IL-2, IL-4, IL-5, IL-6, IL-7, IL-8, IL-10, IL-12, IL-13, and TNF-α.), short-chain fatty acid production, and lipid metabolism were largely unaltered. The fecal *E. coli* loads reduced, with no significant changes to the microbiota [60]. In another trial, the safety of broad-spectrum cocktail, Eliava Pyophage, was tested by comparing the effects of nasal and oral exposure with a mono-species counterpart and placebo in healthy human carriers of *S. aureus*. Physical examination, clinical chemistries, and hematologic studies were analyzed. Fluctuations of body temperature were observed, but none exceeded 38 °C. One had back pain and gastric acidity for 48 h after exposure to the phage cocktail, while two adverse events were noted during exposure to a single phage, with mild pain in the epigastric region for 6 h, and allergic rhinitis and low-grade fever for 72 h, respectively. None of these events were considered by the clinicians likely to be related to the oral phage treatment [62]. In a rhinosinusitis trial, mild adverse events in six patients, including diarrhea, self-resolved epistaxis, symptoms of upper respiratory tract infection, rhinalgia, oropharyngeal pain, or decreased blood bicarbonate level were reported. These were classified as treatment-emergent adverse effects and all resolved without discontinuation of therapy [59].

## 4. Discussion

### 4.1. Potential Impact of Phage Therapy

Humans are exposed to phages in the environment and from their microbiomes. Some studies have suggested that phages can spread into the blood easily and accumulate in distinct tissues [71,72,73,74]. There are even indications that phages are taken up by eukaryotic cells and can trigger innate immune pathways [75,76]. While most of these studies involve temperate, resident phages it is plausible that lytic phages are also able to penetrate eukaryotic cells [77,78]. Nonetheless, the distribution of phages within the body and their impact on tissues and physiologic processes are largely unknown.

#### 4.1.1. Impact of Phage on the Microbiome

The human body harbors a vast and complex microbiome that may contribute to both health and disease [79,80]. The impact of phage therapy on this flora is unclear. In addition, phages are being explored as potential microbial modifiers in infected and microbiota-imbalanced gut disease [57]. A murine model of gut carriage of *E. coli* showed that microbiota diversity was less affected by phage therapy than antibiotics [81]. Two other clinical trials of healthy adults and children also indicated that coliphage reduced the target organism in feces without any considerable change in microbiota composition [60,64]. A pediatric trial of children with diarrheal disease found that oral coliphages transited safely in children with no adverse effects [63]. Additionally, clinical trials with healthy adults and children who ingested coliphage, which targets *E.coli*, showed that fecal phage detection was dependent on the oral dose. No intestinal amplification was detected, suggesting there is passive transit of coliphages through the gut [63,65]. Sarker et al. demonstrated that phage passed through the intestine of healthy people largely passively. Possible adverse effects are limited to the physical presence of virion particles, not to infectious viruses replicating and killing target bacteria. Only when the phage meets its target within patients harboring high numbers of the target *E. coli*, and the mucosal integrity is compromised by the diarrhea pathology, is there concern for the undesirable effects of phage therapy [63].

#### 4.1.2. Endotoxin Release Associated with Bacterial Lysis

Endotoxin is one of the most potent inducers of the inflammatory cytokine response in Gram-negative bacterial infections [82]. As phages can kill bacteria within minutes, phage therapy can potentially result in rapid and significant endotoxin release [83]. There have been a few studies reported regarding potential bacterial lysis-related effects, as mentioned above. However, comprehensive data on the release of endotoxin and its effects are rarely reported and are inconsistent. Endotoxins and other bacterial components that could be present in phage preparations are typically overlooked. These include bacterial DNA [84], Staphylococcal enterotoxin B (a potent bacterial superantigen) [85], alpha hemolysin and other exotoxins [86], or lipoteichoic acid (an important cell wall polymer found in Gram-positive bacteria) [87,88]. Bacterial components and toxins such as endotoxin, which are typically difficult to purify from phage agents, have the potential to induce infusion-related reactions [89,90,91]. These reactions include hypersensitivity and cytokine release syndromes. Symptoms can include flushing, alterations in heart rate and blood pressure, dyspnea, bronchospasm, back pain, fever, urticaria, edema, nausea, and rash [92]. Endotoxin release and infusion-related reactions can be difficult to distinguish, but the presence of these bacterial components should be quantified and documented in phage preparations nonetheless.

#### 4.1.3. Impact of Phages on Immune Activation

Phages have been regarded as bystanders that only impact immunity indirectly via effects on the mammalian microbiome [93]. Recently, both in vitro [94,95] and in vivo [96,97] studies verified that phages also impact innate and adaptive immunity directly [98]. However, results related to immune response instigated by phages are inconsistent and their role in phage therapy is also unclear. Mathematical models have been developed showing their potential importance in a phage therapeutic setting [98,99]. Independent of the phage purification strategy, it is often difficult to attribute these immune responses purely to the phage [96,100].

Phages themselves are immunogenic biological entities that can stimulate adaptive immune responses [101]. Clinical studies in healthy adults, as well as children with acute bacterial diarrhea, showed no detectable phage in the blood stream nor any increase in IgG, IgM, IgA, and sIgA [29,30,46]; however, when administered via intraperitoneal administration, phage triggered increases in phage-specific IgG and IgM antibody titers [19,20]. Phage antibody production may therefore depend on the route of phage administration. In addition, the antibody production was also dependent on the phage type and application time [33,38,70]. Currently, antibody production is thought to affect the efficacy of phage therapy; yet their role in the safety of phage therapy is unclear. Data regarding phage-induced immune responses, including inflammatory cytokine production and antibodies, are an underexplored area and are generally lacking in the studies we reviewed here (Table 1).

### 4.2. Potential Contaminants from Bacterial Components within Phage Preparations

Besides phages themselves, additional components can influence phage safety and toxicity. The major bacterial component that could instigate pro-inflammatory reactions are endotoxins, major components of the cell wall of Gram-negative bacteria that are highly immunogenic in humans [102]. Other potential bacterial components that could be present in phage preparations and are typically overlooked are Staphylococcal enterotoxin B, a potent bacterial superantigen [85], alpha hemolysin and other exotoxins [86], lipoteichoic acid, an important cell wall polymer found in Gram-positive bacteria [87,88], and bacterial DNA [84]. The presence of these bacterial components needs to be quantified and documented in phage preparations.

### 4.3. Potential Chemical Contaminants from Phage Preparation and Purification

Currently there are three major strategies employed regarding the purification of phages. Cesium chloride (CsCl) is often used to obtain high density and high purity phage preparations [91,103]. However, CsCl is typically removed from phage preparations prior to clinical administration as it can be toxic to cells in high concentrations. The most frequently attributed effects of CsCl intoxication are gastrointestinal distress, hypotension, syncope, numbness, or tingling of the lips [104], although a different isotype of CsCl is used in density gradients for phage purification.

Another method of phage purification involves polyethylene glycol (PEG). PEG is an United States Food and Drug Administration-approved biodegradable polymer often used for drug delivery systems [105,106,107,108]. Fortunately, PEG has a high molecular weight and readily undergoes renal clearance leading to a safe toxicity profile and tolerability when used in the phage purification process.

A third method is filtration. Anion exchange is a more controlled purification of phage; however, this method is not ideal for large scale phage purification [109].

### 4.4. The Current Safety and Toxicity Monitoring Associated with Phage Preparations

Table 2 indexes the characteristics of phage preparations described in animal and clinical studies. These characteristics include the phage protein profile, sterility, endotoxin levels, and bacterial DNA levels.

In our review of the literature, data on phage preparations were frequently absent. Almost all studies offered the phage concentration (PFU/mL) directly. Fewer than 40% of the studies reported genotype information. Protein profiles showing the difference between proteins from phage or bacterial origin were mentioned in only 10% of the studies. Twenty-four of the 66 studies described the process used to remove viable bacteria from the phage preparation. Although fewer than 5 units (EU)/kg/hour are required by the FDA in clinical phage preparations [110,111,112], only 14 of the 66 studies reported the level of endotoxin contamination. The bacterial host DNA was reported in only four of the evaluated studies.

Other toxins and contaminations such as lipoteichoic acid, superantigens, or cesium chloride [38,47] were rarely considered in most studies. Additional quality controls regarding shelf life [38,62], pH [41,66], visual appearance [66], were sporadically mentioned. Some phage preparations were developed through commercial production pipelines. Few of these entities reported information regarding phage product manufacturing [17,35,36,37,69,113], although some information on production processes and quality controls was available [114].
viruses-13-01268-t002_Table 2Table 2Characteristics of phage preparations used in the phage therapy studies.ReferenceTitrationCharacterizationComposition & PurityAnimal StudiesPFUGenotypeProtein ProfileSterilityEndotoxinHost Cell DNAOther ToxinsDufour et al., 2019 [15]






Fong et al., 2019 [17]






Drilling et al., 2017 [16]






Drilling et al., 2014 [21]






Chhibber et al., 2008 [115]






Jongsoo et al., 2019 [116]






Chang et al., 2018 [113]






Gelman et al., 2018 [117]






Cheng et al., 2017 [118]






Oechslin et al., 2016 [18]






Galtier et al., 2016 [81]






Jun et al., 2014 [20]






Takemura-Uchiyama et al. 2014 [119]






Osanai, et al. 2012 [120]






Pouillot, et al. 2012 [121]






Ľubomíra Tóthová et al. 2011 [122]






Hung, et al. 2011 [123]






Hawkins, et al. 2010 [124]






Sunagar, et al. 2010 [19]






Nishikawa, et al. 2008 [125]






Case Reports


Lebeaux et al., 2021 [22]






Ferry et al., 2020 [23]






Bao et al., 2020 [24]






Cano et al., 2020 [25]






Rostkowska et al., 2020 [26]






Doub et al., 2020 [27]






Rubalskii et al., 2020 [28]






Gainey et al., 2020 [29]






Aslam et al., 2019 [30]






Nir-Paz et al., 2019 [31]






Tkhilaishvili et al., 2019 [32]






Onsea et al., 2019 [33]






Corbellino et al., 2019 [34]






Susan et al., 2019 [35]






Gilbey et al., 2019 [36]






Law et al., 2019 [37]






RM et al., 2019 [38]






Duplessis et al., 2019 [10]






Kuipers et al., 2019 [39]






LaVergne et al., 2018 [44]






Ferry et al., 2018 [41]






Fish et al., 2018 [40]






Ferry et al., 2018 [45]






Hoyle et al., 2018 [42]






Chan et al., 2018 [43] 






Ujmajuridze et al., 2018 [46]






Schooley et al., 2017 [47]






Zhvania et al., 2017 [48]






Jennes et al., 2017 [49]






Fish et al., 2016 [50]






Fadlallah et al., 2015 [51]






Rose et al., 2014 [52]






Khawaldeh et al., 2011 [53]






Kvachadze et al., 2011 [54]






Letkiewicz et al., 2009 [55]






Clinical Trials






Leitner et al., 2020 [56]






Grubb et al., 2020 [57]






Fabijan et al., 2020 [58]






Ooi et al., 2019 [59]






Febvre et al., 2019 [60]






Gindin et al., 2018 [61]






McCallin et al., 2018 [62]






Sarker et al., 2017 [63]






McCallin et al., 2013 [64]






Sarker et al., 2012 [65]






Rhoads et al., 2009 [66]






Patrick et al., 2018 [67]






Sarker et al., 2016 [68]






Wright et al., 2009 [69]






Dark Blue = Values or result reported within article; Blue = reported, but no specific values or results published within article; Grey = not reported. “Titration” refers to the phage concentration offered by “PFU”. “Genotype” refers to the genetic information, such as the accession number or sequence information of phage. “Protein profile” refers to protein composition of phage; “Sterility” refers to the specific bacterial colony in phage preparation. “Endotoxin” refers to the concentration of endotoxin; “Host cell DNA” refers to the host bacterial DNA; “Other toxins” denotes lipoteichoic acid, superantigens, or CsCl, etc.


### 4.5. Optimization of Safety and Toxicity Monitoring in Phage Therapy

In animal studies, phage doses were variable, ranging from 10^3^ to 10^12^ PFU/ml. None of them defined the median effective dose (ED50), lethal dose for 50% (LD50), or the therapeutic index (TI), a quantitative measurement of the relative safety of a drug that compares the amount of a therapeutic agent that causes the therapeutic effect to the amount that causes toxicity, of the phage preparations. Effects of phage therapy on pregnancy, growth, and development were not described. Additionally, data were mostly limited to rodents and not large animals (e.g., pigs), limiting generalizability to humans. The majority of animal studies utilized IP injection, analogous to IV administration typically used in human studies but challenging to draw direct comparisons.

Clinical safety data analysis and evaluation of new drugs often includes reporting of adverse events, laboratory derangements, changes in vital signs, reviews of systems, and physical examinations of subjects [126]. Biological products such as cytokines, antibodies, and recombinant proteins typically report their immunogenicity. The incidence and consequences of neutralizing antibodies and potential adverse events related to the combination of antibody formation and their adverse reactions were evaluated as well [127]. Including an analysis of the immunogenicity of phages should therefore be an important part of both animal studies as well as case reports. Our review of the phage literature demonstrates the paucity of these data. We believe assessments of safety and toxicity ought to be incorporated into all clinical and preclinical studies of phage therapy, independent of the FDA and the European Medicines Agency (EMA) regulation. Ideally, publications reporting on the safety of phage therapy should include information on the general health of participants, adverse events, chemistry, and hematologic testing data., Information on immune responses should be evaluated prior, during, and after phage therapy. In Table 3, we offer some safety endpoints for consideration that may provide researchers and clinicians guidance on the safety monitoring of phage therapy.

Comprehensive assessments of safety will likely benefit from standardization of safety monitoring. Objective methods of assessment have been employed in some clinical trials, such as gastrointestinal questionnaires or a Visual Analogue Scale (VAS) to assess pain [61,69]. One study utilized a scoring method for assessing physical examination findings in septic mice treated with phage [117]. Another study in a murine bacteremia model introduced a health assessment score [118]. A recent clinical trial applied the National Cancer Institute Common Terminology Criteria for Adverse Events (version 4) to assess the frequency and severity of adverse events during phage treatment [56]. Such methods provide an opportunity to improve safety and the application of scales or standardized scoring methods would better facilitate inter-study comparisons.

In the United States, the Center for Biologics Evaluation and Research (CBER) at the FDA is the main regulatory body overseeing investigational phages [128,129]. The FDA and the EMA mandate that any modern phage therapy products must be made to GMP standards [130,131]. Along with GMP, we feel phage preparations should include information on the characteristics of the phages used in animal studies and clinical studies, including their morphology, genetics, and protein profile, as well as the composition of the phage preparations, including the levels of bacterial contaminants and other impurities. Documentation of the sterility of the phage preparations is necessary. A clear description of the methods used to propagate and purify the phage preparations ought to be provided. These toxicity endpoints are summarized in Table 4.

## 5. Conclusions

There is substantial support for the development of phage therapy as an adjunct to conventional antibiotics. However, proof that phage therapy is safe and non-toxic in humans will be critical for their ultimate success. While phage therapy has generally been safe and well tolerated in studies to date, a comprehensive understanding of the interactions of phage and human hosts is lacking.

Standardized assessments of safety are essential elements of reports of phage therapy in both animals and humans and, although generating these data can undoubtedly be resource intensive, it is ultimately in the interest of all stakeholders engaged in this field to advance this work.

## Figures and Tables

**Figure 1 viruses-13-01268-f001:**
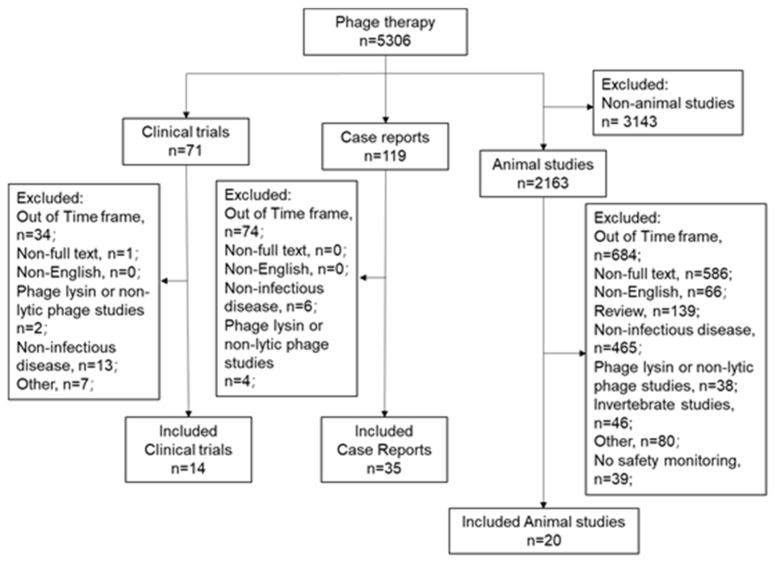
Search strategy.

**Table 3 viruses-13-01268-t003:** Safety endpoints in phage therapy study to be considered.

Safety Monitoring	Safety Endpoints
General assessment	Vital signs; physical exam; subjective symptoms
Labs—Chemistry	Liver function; kidney function; electrolytes; glucose; CRP
Labs—Hematology	CBC with differential; ESR
Pharmacology	Absorption; distribution; excretion; metabolism endpoints (e.g., LE50, ED50, TI)
Immune Response	Non-specific and specific immune responses (e.g., DC, inflammatory factor level; phage specific antibodies)

Abbreviations: Erythrocyte sedimentation rate (ESR); C-reaction protein (CRP); CBC: Complete Blood Count; WBC: White blood cells; DC: CBC with differential; BPC, Blood platelet count; LE50, Lethal Dose 50; ED50, Median Effective Dose; TI: Therapeutic Index.

**Table 4 viruses-13-01268-t004:** Characteristics of phage preparation to be considered.

Phage Parameters	Phage Preparation Measurements
Identify	Morphology
Potency	Titer
Sequencing	Genotype; Protein profile
Bacterial contaminants	Viable bacteria; Endotoxin; Enterotoxin B; Bacterial DNA
Other impurities	CsCl
Others	Sterile; PH; shelf time; suspended buffer; osmotic pressure

The morphology, titration and genomic description of the used phage, including the genome sequence as well as a complete annotation of the proteins encoded in the genome. The presence of both bacterial remnants, endotoxin level, bacterial DNA, as well as potential presence of toxic components of the purification method itself; Sterility, suspended buffer, pH stability, temperature range and shelf life should be denoted.

## Data Availability

Not applicable.

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
