# Peer review of "The Safety and Toxicity of Phage Therapy: A Review of Animal and Clinical Studies"

_viruses, 2021, doi:10.3390/v13071268_

Round 1

Reviewer 1 Report

This work data about monitoring of safety and toxicity of phage therapy, due to the prioritization of the advantage’s description of these types of therapies found in bibliography available. However, although the focus of this review is very interesting and would contribute with valuable recovered information, I found the scope missing in most of the manuscript since it is not really focused on the safety and toxicity from the beginning of the text, so I recommend reconsidering its acceptance under major revision. The major concerns found are described below.

Introduction:

  • Although the authors mention the intention of this work, the state of the art provided is scarce or non-existent regarding the topic that is going to be discussed in the article, so it does not allow to focus the topic of the work (all that information is placed only in the Discussion).

Materials and Methods:

  • Figure 1. Although the excluded cases are described in the manuscript, looking at the Figure 1 is very confusing. All the excluded cases are mentioned but no indication of which of them are excluded into the figure. I suggest clarifying this matter into the figure and/or the title of the figure.

Results:

  • Line 60-61, Although it can be understandable, when mentioned “pre-clinical studies” should be clarified as in the Table 1 “Animal studies”.
  • Although the study is said to be focused on the safety and toxicity of the phage therapy, in most of the cases, authors focus their manuscript on the efficiency of the therapy applied. For example, when analysing the study of Dufour et al., authors did not even mention that that study did not analysed the adverse events found.
  • Moreover, when adverse events were found, it is very relevant to describe how many adverse events and their intensity (if it is available) in regard of the total of the sample used in the study, not only a table with the type of study carried out, to analyse their frequency. Resuming, an analysis of this topic should be included if the scope of the manuscript is safety and toxicity of phage therapy, not only a description of the cases found.
  • I found very interesting supplementary figure 2 since it allows to compare the frequency of the administration routes in every type of therapy and compare it with the adverse effects (even when I could not elucidate whether they could be correlated, could they?). Thus, I suggest including this figure as a standard figure instead a supplementary one and develop an approach to a possible correlation with the adverse effects if possible.

Discussion:

  • If the scope of the manuscript is the safety and toxicity in phage therapy, the most relevant information recovered is from point 4.5. Safety and Toxicity Associated with Phage Preparations onwards. In this sense, it would be interesting to focus the manuscript on those points and discuss and compare with the advantages (immune response and so on) of this type of therapies. On the contrary, when reading, the focus is most of the time lost in the information described.

Reviewer 2 Report

The approach of this review is interesting and innovative given the use of Phage therapy, which is often restricted due to the lack of regulatory guidelines and the lack of unknown data of phage application in humans. Therefore, it is important to explore and demonstrate the safety/toxicity of phages in clinical applications.

I consider this manuscript suitable for publication, after the authors address the minor issues outlined below:

  • Line 22: ”…to optimize” instead of “…to optimizing”.
  • Line 44: “in vivo” should be written in italic.
  • Lines 54-57: The data of the following sentence “The time frame was defined as January 1, 2008 to March 28, 2021. Full text and English-language articles were included; Non-research articles, reviews, opinion pieces were excluded. Studies involving invertebrates, non-infection diseases, phage lysins, or non-lytic phages were excluded.” is redundant because the sentence just before Figure 1 states the same.
  • Line 61: “et al.” in italic. Line 72 and 197 also.
  • Line 64: because this is the first mention to coli please write Escherichia coli, the following mentions you can write E. coli.
  • Line 65: please refer which antibiotics were used.
  • Line 71: why put IFN-γ between brackets and IL-12 not?
  • Line 72: because this is the first mention to aureus please write Staphylococcus aureus, the following mentions you can write S. aureus including in line 77.
  • Line 78: EDTA alone was also used? And the same results were observed alone and combined with CTSA? If so, EDTA can be used without phages. I will recommend not to mention to this study.
  • Line 84: because this is the first mention to aeruginosa please write Pseudomonas aeruginosa, the following mentions you can write P. aeruginosa.
  • Line 91: Sixteen instead of 16. In line 111 you start the sentence with Thirty-five. Be consistent.
  • Line 100: Vibrio parahaemolyticus and aureus in italic.
  • Line 128: because this is the first mention to baumannii please write Acinetobacter baumannii; after intravenous put (IV) because in line 147 you just write IV.
  • Line 137: “who experienced” or “experiencing”.
  • Line 138: aeruginosa in italic.
  • Line 142: baumannii.
  • Line 160: coli in italic.
  • Line 165: aureus in italic.
  • Line 189: coli in italic and line 201 also.
  • Line 195: coli in italic and space between E. and coli.
  • Line 203: remove the full, stop before the reference.
  • Line 223: Phages not Phage.
  • Line 225: verified.
  • Line 234: however not However.

Round 2

Reviewer 1 Report

I have to thank the authors for the appropiated change of organization and for giving this new version a more concrete focused in toxicity and safety regarding phage therapy. Emphasizing the lack of information in this therapy is very useful to be able to improve it. 

This new version of the manuscript y more focused as in the introduction they go straight to the point and remark the relevance of the issue although one of the major points is the existence of gaps in the "negative" aspects of this type of therapy, is clearer and more understandable so I suggest to accept it. 

Author Response

Thank you kindly for the favorable review.  The manuscript is much stronger as a result of your feedback.